# Effort-Reward Imbalance, Over-Commitment and Depressive Episodes at Work: Evidence from the ELSA-Brasil Cohort Study

**DOI:** 10.3390/ijerph16173025

**Published:** 2019-08-21

**Authors:** Tânia Maria de Araújo, Johannes Siegrist, Arlinda B. Moreno, Maria de Jesus Mendes da Fonseca, Sandhi M. Barreto, Dóra Chor, Rosane Härter Griep

**Affiliations:** 1Health Department, State University of Feira de Santana, Bahia 44036-900, Brazil; 2Institute of Medical Sociology, Centre for Health and Society, Medical Faculty, University of Dusseldorf, 40225 Düsseldorf, Germany; 3Department of Epidemiology and Quantitative Methods in Health, National School of Public Health Sérgio Arouca, Fundação Oswaldo Cruz, Rio de Janeiro 21041-210, Brazil; 4Faculty of Medicine, Universidade Federal de Minas Gerais, Belo Horizonte, Minas Gerais 30130-100, Brazil; 5Laboratory of Health and Environment Education, Oswaldo Cruz Institute, Oswaldo Cruz Foundation, Rio de Janeiro 21040-900, Brazil

**Keywords:** Brazil, work stress, effort-reward imbalance, over-commitment, depressive episodes, ELSA-Brasil

## Abstract

A growing burden of mental illness, and in particular depression, among workers is a concern of occupational public health. Scientific evidence has revealed consistent associations of work-related stress, as measured by theoretical models, with depression, but mostly so in developed countries. This contribution explores these associations in a developing Latin American country, Brazil, by applying an internationally established work stress model, the effort-reward imbalance (ERI). This model focuses on the work contract where unjust exchange between high efforts spent and low rewards received in turn contributes to stress-related disorders. The model’s extrinsic (‘effort’, ‘reward’) and intrinsic components (‘over-commitment’), as well as their combination, are hypothesized to be related to a higher risk of depressive episodes (DE). Using cross-sectional data from the ELSA-Brasil study, including 10,034 workers from the public sector, we observed increased prevalence ratio (PR) of DE according to ERI scales. The quartiles of highest ‘effort’ (PR = 1.85; 1.44–2.37), highest ‘over-commitment’ (PR = 3.62; 2.80–4.70) and lowest ‘reward’ (PR = 3.44; 2.55–4.64) were associated with DE, on adjusted models, as well was the E–R ratio (PR = 2.47; 1.92–3.17). An additive interaction was identified between the E–R ratio and ‘over-commitment’. The results support the use of ERI as a screening tool for work stress in the Brazilian context and will offer guidance for worksite health promotion programs.

## 1. Introduction

The nature of work and employment has undergone significant changes in recent decades in high income and developing countries. With the advent of new technologies and rising competition in the frame of a globalized economy, employment became more flexible and less physically demanding, but at the same time less secure and more stressful in terms of high mental and emotional workload and in terms of threats to job continuity and promotion prospects [1,2]. These changes, if experienced recurrently, can adversely affect the health and wellbeing of working people. In fact, based on the availability of distinct stress-theoretical concepts, a variety of prospective epidemiological investigations have confirmed this assumption, by demonstrating elevated risks of incident stress-related disorders, such as depression [3,4], and ischaemic heart disease [5,6], among exposed as compared to unexposed people. In this context, two theoretical models of stressful work have been analyzed with special intensity, ‘demand-control’ and ‘effort-reward imbalance’. The former model claims that stressful experience results from jobs that fail to offer control and decision latitude while putting high quantitative demands (e.g., work pressure) on working people [7]. As a complementary theoretical concept, effort-reward imbalance is concerned with stressful features of the work contract, with a focus on social reciprocity in costly transactions. Failed reciprocity in terms of high effort spent and low reward received in turn elicits strong negative emotions and stress reactions. In this model, three reward transmitters are distinguished (money, esteem, social status), and the model includes an intrinsic component (personal coping with work in terms of over-commitment) in addition to its extrinsic components [8].

Epidemiological evidence on associations of stressful work with health was mainly derived from Western high-income countries (Europe and North America). It is of interest to know whether these findings hold true in developing countries, and in particular in the context of a Latin American country like Brazil. In Brazil, stressful working conditions have become an issue of growing public concern more recently, given the rapid transformation of the labor market with its large amount of informal jobs and insecure working conditions, based on the social precariousness of labor, i.e., a process of economic, social, and political institutionalization of flexibilization [9]. With this paper, we set out to analyze associations of stressful work, as measured by an effort-reward imbalance, with an important indicator of mental health, depressive episodes, in a large population of employed men and women.

Our focus on mental health is justified by its significance in terms of population health, as evidenced by the increase in the prevalence of mental disorders related to work, especially the growth of cases of depression [10,11,12], and economic costs due to absenteeism, work dismissal and early retirement [13,14]. For the following reasons we measured work stress by the effort-reward imbalance (ERI) model. First, with its emphasis on job insecurity, lack of promotion prospects and poor pay it captures crucial aspects of currently stressful work in this country. Second, as this model distinguishes between an intrinsic and extrinsic dimension of stressful experience it allows us to analyze the relative contribution of situational and personal aspects towards explaining mental health at work. Importantly, persons scoring high on over-commitment are highly susceptible to mental stress due to their excessive engagement and a desire of being in control [15]. Assessing both extrinsic and intrinsic components of the model has direct implications for prevention at work. Third, given the large-scale measurement of the original version of the ERI questionnaire in Portuguese, the model’s performance in the Brazilian context, especially its ability to discriminate different groups of occupational exposures and its effect on health, can be analyzed.

More specifically, the following hypotheses are tested: 1. Each model component (high effort, low reward, high over-commitment) is associated with elevated prevalence ratios (PR) of depressive episodes (DE). 2. Prevalence ratios of depressive episodes related to the ratio of effort and reward are higher than prevalence ratios related to the single model components effort or reward respectively. 3. Over-commitment moderates the association of the effort-reward ratio with depressive episodes. The second and third hypotheses deserve a further comment. As the ratio of effort to reward quantifies the amount of failed reciprocity at an individual level, it may indicate an additional source of stressful experience. Finally, the third hypothesis claims that the strength of associations of effort and reward with mental health indicators is moderated by a personal pattern of coping, defined as excessive striving (over-commitment).

## 2. Materials and Methods

### 2.1. Study Design and Population

This study analyzed data from ELSA-Brasil—Brazilian Longitudinal Study of Adult Health. ELSA-Brasil is a large-sized multicenter longitudinal study that involves public education and research institutions in six Brazilian capitals (Belo Horizonte, Porto Alegre, Rio de Janeiro, Salvador, Sao Paulo, Vitória). It is a broad-scope study and is considered to be one of the largest ones so far developed in the field of epidemiology in Latin America. The study population consists of public-sector workers at the six institutions involved. The cohort includes professors and researchers; clerks; secretaries and other office workers; nurses (registered, assistant, and licensed practical); medical and laboratory technicians; and installation, maintenance, and repair workers. Sampling wave 1 took place between August 2008 and December 2010; wave 2 took place between 2012 and 2014. The baseline was composed of 15,105 volunteers: men and women, who were either actively working or were retired, with ages between 35 and 74 years. In the second wave, 14,014 workers (active and retired workers) participated. The study design and sampling procedures of ELSA-Brasil have been reported previously [16] (more information can be found in: Revista de Saúde Pública, vol. 47, supplement 2, which is dedicated to methodological aspects of the ELSA-Brasil—an English version was available (Revista de Saúde Pública. Available online: http://www.scielo.br/scielo.php?script=sci_issuetoc&pid=0034-891020130008&lng=en&nrm=iso).

In this study we include only active workers who participated in the second wave (N = 10,034 workers). The Elsa-Brasil sample included similar proportions of both genders, as well as predefined proportions of specific age groups and distinct occupational categories, to permit a wide socioeconomic gradient.

ERI data are currently available from the second wave of ELSA-Brasil, when this measurement was included (2012–2014). Therefore, although this is a cohort study (follow-up), the data analyzed here are based on a cross-sectional design, referring to a single measure in time. 

### 2.2. Measurement of Work Stress

In this study, occupational stressors were evaluated by the Effort-Reward Imbalance (ERI) Model. The original version of ERI contains 23 items distributed in three scales: effort (six items), reward (11 items) and over-commitment (six items). ERI items are measured on a Likert-type response scale ranging from one to four (Strongly Disagree, Disagree, Agree, Strongly Agree).

In the composition of the effort scale, according to the characteristics of the population studied (most of the population studied (82%) developed non-manual activities, not involving significant physical demands), the item ERI_5 “My job is physically demanding” was excluded. This procedure is in agreement with guidelines of use of the ERI model [17]. Thus, the effort scale contained five items, ranging from 5 to 20 points. The reward scale is composed of three subscales: esteem (five items), job promotion (four items), and job insecurity (two items), totaling 11 items. In Elsa-Brasil, on the job insecurity scale, item ERI_13 “My employment security is poor” was excluded because the ELSA-Brasil population consists of civil servants with a stable employment relationship. Thus, the reward scale included 10 items, ranging from 10 to 40 points. The over-commitment scale included 6 questions, ranging from 6 to 24 points.

The score of each component was calculated, with the higher scores reflecting higher effort, higher reward, and higher over-commitment. In addition, an effort-reward (ER) ratio was analyzed, defined by the sum of the effort score, divided by the sum of the reward score, multiplied by the correction factor that accounts for the unequal number of items [17]. This ratio quantifies the extent of mismatch between effort spent and reward received at individual level. In the original proposition of the ERI Model, a ratio of 1.0 is supposed to reflect a balanced experience, ratios exceeding this threshold indicate a stressful experience of perceived injustice of exchange. This imbalance, in turn, can produce adverse effects on physical and mental health. However, it is noteworthy that more recently it has been proposed to analyze this imbalance (E–R ratio) based on the distribution obtained in the investigated samples, recommending, preferably, the use of continuous data analysis, instead of the specific threshold at 1, allowing the exploration of data in quartiles, for example [18]—procedure we followed in this study. 

### 2.3. Measurement of Mental Health

The Clinical Interview Schedule-Revised (CIS-R) was the instrument used to assess mental health [19]. CIS-R is a structured interview for the measurement and diagnosis of non-psychotic, psychiatric morbidity developed by Lewis et al. [20], for use in community surveys. Several studies have reported its validity and reliability when applied to different countries and settings [21]. The original version of the CIS-R was translated and adapted to Brazilian Portuguese and then back-translated, and analysis of conceptual equivalence between items, semantics, and operational part was performed [22,23].

CIS-R allows the following diagnostic categories of ICD-10: generalized anxiety disorder, depressive episodes, all phobias (agoraphobia, social phobia, and simple phobia), obsessive-compulsive disorder, panic disorders, and mixed anxiety and depressive disorder. In this study, our outcome is depressive episodes.

In CIS-R, information on the categories of disorders occurs in two stages. First, informants’ responses to CIS-R were used to produce specific diagnoses based on ICD-10 [20,22]. This procedure is done by applying algorithms, according to standardized computer procedures for CIS-R. After that, ICD-10 diagnoses were grouped to establish categories. The CIS-R algorithms regarding depressive episodes include items related to F32.00, F32.01, F32.10, F32.11, and F32.20. The categories include mild or moderate depressive episodes (with or without somatic symptoms) and severe depressive episodes [24]. In this study, depressive episodes was dichotomized, grouping all types of depressive episodes (mild to severe), compared to participants without any kind of depressive episodes.

### 2.4. Measurement of Covariates

In addition to the variables of interest (exposure and outcome) we included the following covariates: a) Age—measured in years and analyzed in five-years age bands (≤40 years, 41 to 45; 46 to 50 years, 51 to 55 years, 56 to 60 years and more than 61 years); b) gender (female, male); c) marital status (single, married/living together, divorced/separated/widower); d) self-identified skin color/race category (white, brown [mixed], black, yellow, and indigenous), according to the categories used in the National Census; e) educational level (incomplete elementary school/complete elementary school (Middle school), complete secondary school (High school), and university degree (undergraduate and master/PhD degree).

### 2.5. Statistical Analysis

Sociodemographic characteristics were described using frequencies and percentages. The analysis of the association between ERI dimensions and depressive episodes was developed in stages, in line with our working hypotheses. To evaluate whether each model component (high ‘effort’, low ‘reward’, high ‘over-commitment’) is associated with the prevalence of depressive episodes we first calculated the score distribution of the ERI scales, and defined quartiles of their distribution. Subsequently, the prevalence of depressive episodes, the prevalence ratios and the respective 95% confidence intervals were estimated for each scale. Crude, age-adjusted and fully adjusted models were tested. The analysis of dose-response effect was based on the Wald test. The Poisson regression model with robust variance was used to estimate prevalence ratios [25]. Essentially the same analysis was performed with regard to the second hypothesis, where quartiles of the E–R ratio were constructed. 

For the evaluation of the third hypothesis, the interaction of ‘over-commitment’ and the E-R ratio with depressive episodes, we tested both multiplicative and in the additive scales. In the first case, we tested interaction with the inclusion of a product term (multiplicative model). To this end, the E–R ratio and ‘over-commitment’ variables were dichotomized (the highest quartile as exposure vs. remaining quartiles). A Poisson regression was performed, including the product term in the adjusted model. In the second case, interaction was studied according to the criterion of departure from additivity (additive model). We introduced a dummy variable based on the two dichotomized measures (highest quartile of the E–R ratio; highest quartile of ‘over-commitment’), with four categories: (1) low E–R ratio, low ‘over-commitment’ (00), (2) high E–R ratio, low ‘over-commitment’ (10), (3) low E–R ratio, high ‘over-commitment’ (01), (4) high E–R ratio, high ‘over-commitment’ (11). 

The measures of interaction based on the additivity criterion were verified by calculating the excess risk due to interaction (RERI = PR11 − PR01 − PR10 + 1), which quantifies the deviation from the null value, and the attributable proportion due to interaction (attributable proportion due to interaction – AP = (PR11 − PR01 − PR10 + 1)/PR11), which shows the proportion of cases from both exposures, and synergy index (S = (PR11 − 1)/(PR01 + PR10 − 2)) that reflects the direction of the interaction in relation to nullity (S = 1), synergy (S > 1), or antagonism (S < 1) [26]. We also calculated excess of prevalence (EP = P_exposure_ – P_no exposure_), which indicates whether the combined effect of the factors is greater than the sum of their individual effects [27,28,29]. To calculate the additive interaction estimates and respective 95% confidence intervals, the procedures proposed by VanderWeele and Knol [28] were followed.

All analyses were performed using STATA 12.0 (StataCorp LLC, Texas, United States of America). 

### 2.6. Ethical Approval

Considering that ELSA-Brasil is a multi-center study, the project was approved by the Research Ethics National Committee (Comitê Nacional de Ética em Pesquisa) and by the committees of each institution involved in December 2008 (Study registration number: 140/08). The volunteers gave written consent to participate.

## 3. Results

### 3.1. Descriptive Findings

The second wave of ELSA-Brasil included 14,014 participants. Of these, 10,034 were active, working in the participating institutions of the cohort. The proportions of men and women were relatively similar, although with 3.4% more women (Table 1). More than half (67.2%) were married or living with a partner; people who declared themselves whites slightly predominated (51.7%) compared to brown and black people. There was also a high educational level: 49.3% had higher education, 17.4% of whom had a master or Ph.D. degree. The mean age was 52 years (minimum of 38 and a maximum of 71 years).

### 3.2. Results Related to the Hypotheses

The prevalence of depressive episodes was 4.8%. We observed an increase in the prevalence of depressive episodes in line with quartiles of the ERI scales analyzed, in a dose-response gradient (Table 2). The prevalence ratios of depressive episodes increased according to the quartiles of ‘effort’ (i.e., higher prevalence ratios with higher ‘effort’, lower ‘reward’, higher ‘over-commitment’, and higher ‘effort’-‘reward’ ratio). These findings support the first research hypothesis. Concerning the second hypothesis (that the prevalence ratios of depressive episodes related to E–R ratio are higher than the prevalence ratios compared to the single model components effort or reward), confirmation was restricted to the scale ‘effort’, while the prevalence ratio of low ‘reward’ exceeded the one of the variable combining the two components. With a PR of 3.75 the lowest quartile of ‘reward’ documented a very strong association that was above all other estimates.

Very similar findings resulted from the analysis of adjusted models (age-adjusted and fully adjusted) (Table 3). Again, the prevalence ratio of low ‘reward’ was higher than the one of the E–R ratio. Interestingly, ‘over-commitment’, the personal component of this theoretical model, was very strongly associated with depressive episodes. 

Finally, to evaluate the hypothesis of an interaction between E–R ratio and ‘over-commitment’, a multiplicative and an additive model were analyzed. Table 4 shows the results of the multiplicative model test. While both variables showed independent associations with depressive episodes, based on the fully-adjusted model, the term-product did not reach the required level of statistical significance. Therefore, this approach did not support the hypothesis. 

In Table 5, the findings of the additive model are displayed. The analysis of the effect of isolated factors showed that workers with high E–R ratio had higher prevalence of depressive episodes than the reference group (1.71; 1.29–2.25), as well as those who reported high ‘over-commitment’ (2.02; 1.53–2.68). Among those exposed to both exposures, we could perceive greater magnitude of association (3.68; 2.97–4.56).

The difference between the expected prevalence for combined exposure (P = 8.25%) and the prevalence observed for the group (P = 10.79%) exceeded the occurrence of the outcome in 2.54%. The association of the combined effect of high E–R ratio and high ‘over-commitment’ was higher than what was estimated based on the additivity of the effects, resulting in RERI = 0.95. The attributable proportion due to interaction was 25.0% with direction for the synergy of the effects (S = 1.54). 

The evaluated indicators show the existence of cases resulting from the joint exposure to the analyzed factors (high E–R ratio and high ‘over-commitment’). That is, the observed prevalence was higher than expected by the combination of cases in each isolated exposure situation. Similar result was found for the prevalence ratio (PR), higher than PR due to the expected combined effect. Therefore, hypothesis 3 was supported by the results of additive model analysis.

## 4. Discussion

This study demonstrated consistent associations of stressful work, as measured by ‘effort’-‘reward’ imbalance, with depressive episodes in a large population of Brazilian men and women employed in the public sector (the ELSA-Brasil study). Significantly elevated prevalence ratios of depressive episodes were observed for each model component, i.e., high ‘effort’, low ‘reward’, and high ‘over-commitment’, as well as for a combined measure quantifying the ‘effort’-’reward’ imbalance. A particularly strong association was found for participants who scored in the lowest quartile of ‘reward’, exceeding even the association of the combined ‘effort’-‘reward’ ratio with depressive episodes. ‘Over-commitment’ was equally strongly related to depressive episodes. Additional evidence in favor of an interaction of the ‘effort’/‘reward’ ratio and of ‘over-commitment’ with depressive episodes was observed in an additive, but not in a multiplicative interaction model. 

These cross-sectional findings are in line with previously reported evidence derived from prospective cohort studies, where ‘effort’-‘reward’ imbalance at work was associated with a 1.5-fold increased risk of depression [3]. Importantly, the current results are among the few ones resulting from a Latin American country, whereas most of this research was generated in economically developed countries in Europe and North America. As already mentioned, Brazil is currently subjected to a rapid transformation of the labor market that is not restricted to the employment relationship, but also affects other important aspects of work life, such as deregulation of labor legislation, occupational safety and health, and reduced impact of trade unions [9,30]. Thus, the associations of stressful work with depressive episodes are of particular interest, specifically so as the ERI model captures some important aspects of precarious work (low promotion prospects, poor pay, lack of recognition). The relationship between ‘over-commitment’ and depressive episodes represents a further study outcome that deserves attention. The ERI model is one of the very few theoretical concepts that includes a personal component of stressful experience in addition to the situational component(s), thus allowing a quantitative estimate of the relative contribution of these components towards explaining a health indicator under study. A previous review confirmed that the consistency of associations of ‘over-commitment’ with a range of health indicators is comparable to the one observed for the extrinsic model components [15]. The same review also included the results of some studies that showed an additional interaction of this intrinsic component with the extrinsic ones, a finding that is in line with the observed additive interaction effect reported in Table 5. These findings support the notion that person-based aspects of work stress deserve attention in worksite preventive activities in addition to organizational-structural attempts of reducing effort and increasing reward. 

Distinct from previous research (for review Siegrist and Wahrendorf [31]), the current study did not support the notion that the strength of a combined measure of high ‘effort’ and low ‘reward’ would exceed the strength of their separate measures, given the remarkably strong association of critically low ‘reward’ with depressive episodes. It is possible that the stressful effects of high ‘effort’ are less pronounced than those of low ‘reward’ in this cohort of public employees (see also the relatively low prevalence ratios of depressive episodes in case of high ‘effort’ in Table 3). In additional sensitivity analyses we explored associations of the sub-components of the ‘reward’ scales with depressive episodes. Interestingly, all three subscales revealed relationships of similar strength, confirming the relevance of material components (low promotion prospects, exposure to undesirable changes) as well as psychosocial components (lack of esteem and support) of low occupational rewards (Appendix A). Clearly, the strong association of low ‘reward’ with poor mental health deserves further analyzes, with a special focus on labor market and employment conditions of working people in Brazil.

A final aspect to be highlighted concerns the interaction analysis between ERI scales. We tested multiplicative and additive interaction, as has been most recently recommended [28]. Multiplicative interaction analysis predominates in the literature, including studies that explore the hypothesis of an interaction between ERI dimensions and health effects [15]. This is mainly due to easy access to procedures for estimating this type of analyzes in statistical packages, a fact that less obvious in case of testing additive interaction. In our case, additive interaction analysis is particularly relevant as it allows to assess the impact of work stress on specific subpopulations. This information can instruct prioritization of intervention efforts including cost-effectiveness [26,29]. In this study, we found an additional interaction between ‘effort’-‘reward’ ratio and ‘over-commitment’ for the outcome of depressive episodes. Different estimated indicators were calculated, confirming such an interaction. 

### Strengths and Limitations

This study has several strengths. First, its results are based on a large, relatively homogenous population from one of the largest cohort studies in this Latin American country, the ELSA-Brasil study. Although the study population is not representative of the Brazilian working population (employment in public sector; high educational level; majority of white population, high mean age), it allows a differentiated analysis according to gender, age, marital status, or race/color. For instance, it is of interest to note that behavior of the associations of stressful work with depressive episodes were very similar among female and male employees, despite a higher prevalence of depressive episodes among women (see Appendix A). A second strength concerns the application of validated measures of the two core sets of variables under study, stressful work and depressive episodes. Stressful work was assessed by psychometrically validated scales of a theoretical model that has been widely applied in international research [31,32]. In this study, the internal consistency of the scales and their factorial structure have been tested and confirmed again, but a detailed presentation and discussion of these findings would exceed the scope of this contribution and will be given in a separate publication. Similarly, the measurement of depressive episodes was based on a validated Brazilian Portuguese version of the Clinical Interview Schedule-Revised (CIS-R), a structured interview assessing clinically relevant mental health problems for use in community surveys [22]. Out of six ICD-10 diagnostic criteria we focus on the category of depressive episodes, due to its important contribution to the burden of mental health in working populations [10,11,12]. The elaborated statistical analyzes of available data are considered a further strength of this study. We tested the hypotheses of single and combined associations of work stress components with depressive episodes with multivariable models, adjusting for relevant confounders, and we applied two formal approaches towards testing the interaction hypothesis, a multiplicative and an additive statistical model.

Yet, the study also has some limitations. The cross-sectional design of the analysis due to the availability of the ERI scale only in the second wave interview prevents us from assumptions of directionality of the effects. In addition, we cannot exclude a systematic bias due to altered responses to the work stress items by participants who suffered from depressive episodes at the time of data collection, which would suggest reverse causation [33]. Considering that the study population was composed by active workers attenuates the effect of mild/severe depressive episodes. Besides this, the fact that prospective associations of work stress with depression were confirmed in other studies indicates that the likelihood of reverse causation may be relatively small [3,31,34]. However, despite pathophysiological evidence of an impact of social reward deficiency on the risk of developing depression it is possible that depressed people experience and perceive rewarding conditions at work as less meaningful and beneficial. 

Related to the cross-sectional design, a second limitation is given by the methodological problem of common method variance [35]. This problem arises from the fact that both the ‘exposure’ and the ‘outcome’ constructs were assessed by self-reported data. However, both measures, the CIS-R data and the ERI questionnaire, were derived from structured interviews and psychometrically validated scales respectively, and applied researcher-based scores, thus minimizing reporting bias. Thus, the measurement instruments employed were based on standardized assessments. It should be highlighted that, in our case, as cross-sectional findings were replicated by prospective results, this limitation does not seem to have a major impact [3]. 

Third, our findings cannot be generalized to the working population in Brazil, given the study’s specific sample criteria, nor can they be generalized to public sector employees, due to the focus on public educational and research sectors of employment and the restriction to employees in urban centers. Yet, in view of the size of the cohort under study, and in view of the consistency and strength of associations, the findings are of general interest as they reveal a strong relationship between poor quality of work and poor mental health even in a relatively privileged part of employed people in this Latin American country. Finally, the scope of analysis of the present study is limited as only one out of several work stress models has been studied, and as only one mental health dimension has been analyzed. Important stress-related factors, such as work-life conflicts or conditions of financial hardship were not included. The same holds true for important contextual factors, such as toxic work environments or conditions of socio-political and economic instability. However, the study findings extended available evidence on work-related health beyond Western developed countries by exploring its role in an important developing country.

In spite of these limitations, the results found in this study are especially relevant, considering that the Latin American literature on the association between occupational stressors and mental health is scarce. Elsa-Brasil is a multicenter study, a cohort involving a large sample of the working population (more than 10,000 active workers)—not a record of another study of similar magnitude in the region—and offers a considerable amount of information rarely available in Brazil. Moreover, this knowledge is relevant in policy terms, offering new insights into targets for health-promoting activities at work. Several intervention studies based on this theoretical model demonstrated beneficial effects on mental health. For instance, in a Canadian study [36], employees in five organizations with implemented improvements of leadership and recognition at work exhibited a significant reduction of low reward at work and of psychological distress over a four-year follow-up period, compared to employees in five organizations without this intervention. 

## 5. Conclusions 

This study demonstrated consistent associations of stressful work, as measured by the ‘effort’-‘reward’ imbalance, with depressive episodes in a large population of Brazilian men and women civil servants (the ELSA-Brasil study). Low ‘reward’ at work and ‘over-commitment’ were strongly related to this outcome. Moreover, an additional interaction of the extrinsic and intrinsic components of the model was observed. These Brazilian findings along with all the previously published literature about this topic in developed countries might be useful for instructing programs on worksite mental health promotion. Such programs are particularly timely in Brazil, a country undergoing rapid and stressful labor market changes.

## Figures and Tables

**Table 1 ijerph-16-03025-t001:** Sociodemographic characteristics of working population in the ELSA-Brasil study, 2012–2014.

Characteristics (N)	*n*/Mean	%/SD
Age (N = 10,034) *	52.06	±6.7
Gender (N = 10,034)		
Men	4847	48.3
Women	5187	51.7
Marital Status (9947)		
Single	1346	13.5
Married/Living together	6685	67.2
Divorced/Separated	1611	16.2
Widowed	305	3.1
Race/skin color (9920)		
Black	1590	16.0
Brown	2876	29.0
White	5124	51.7
Yellow (Asian descents)	232	2.3
Indigenuos	98	1.0
Education (N = 10,032)		
Elementary/Middle school (till 8 years)	649	6.5
High school (3 years)/Incompleted undergraduate	3438	34.3
University (undergraduate)	3201	31.9
Master degree	800	8.0
Doctoral degree	1944	19.4

* Age: Minimum: 38 years; Maximum: 71 years.

**Table 2 ijerph-16-03025-t002:** Prevalence of depressive episodes (%) and crude associations of the ERI scales with depressive episodes. ELSA-Brasil Study, 2012–2014.

Scales (N)/Quartiles	Prev. (%)	PR	95%CI	*p* Value *
Global prevalence (10,018)	4.8			
Effort (9987)				
Quartile 1	3.85			
Quartile 2	4.72	1.27	0.94–1.59	
Quartile 3	5.01	1.30	1.02–1.64	<0.001
Quartile 4	6.29	1.63	1.28–2.08	
Reward (9619)				
Quartile 1	8.53	3.75	2.79–5.04	
Quartile 2	3.92	1.72	1.22–2.43	
Quartile 3	3.56	1.55	1.09–2.19	<0.001
Quartile 4	2.27	-	-	
Over-commitment (9992)				
Quartile 1	2.85	-	-	
Quartile 2	3.14	1.10	0.81–1.48	
Quartile 3	5.26	1.84	1.38–2.46	<0.001
Quartile 4	8.61	3.01	2.34–3.87	
E-R ratio (9534)				
Quartile 1	3.62	-	-	
Quartile 2	3.45	0.95	0.70–1.28	
Quartile 3	3.96	1.09	0.82–1.45	<0.001
Quartile 4	8.38	2.31	1.81–2.95	

* Wald Test was estimated to analysis of dose-response effect (prevalence of the depressive episodes according to ERI scales). Prev. = prevalence rate; PR = prevalence ratio; CI: Confidence Interval.

**Table 3 ijerph-16-03025-t003:** Association between ‘effort’-‘reward’ imbalance and depressive episodes including prevalence ratio (PR) and respective 95% confidence intervals. ELSA-Brasil, 2012–2014.

Scales	Model 1 *		Model 2 **	
	PR	95%CI	PR	95%CI
Effort				
Quartile 1	1.00	-	1.00	-
Quartile 2	1.21	0.93–1.57	1.22	0.94–1.59
Quartile 3	1.27	1.01–1.61	1.36	1.08–1.73
Quartile 4	1.62	1.27–2.06	1.85	1.44–2.37
Reward				
Quartile 1	3.67	1.0–2.17	3.44	2.55–4.64
Quartile 2	1.71	1.21–2.41	1.64	1.16–2.31
Quartile 3	1.53	2.72–495	1.51	1.06–2.13
Quartile 4	1.00	-	1.00	-
Over-commitment				
Quartile 1	1.00	-	1.00	-
Quartile 2	1.10	0.82–1.49	1.17	0.87–1.59
Quartile 3	1.86	1.40–2.49	2.14	1.59–2.88
Quartile 4	3.05	2.37–3.92	3.62	2.80–4.70
E-R ratio				
Quartile 1	1.00	-	1.00	-
Quartile 2	0.94	0.93–1.27	0.99	0.73–1.33
Quartile 3	1.08	0.92–1.27	1.17	0.88–1.57
Quartile 4	2.27	1.63–2.14	2.47	1.92–3.17

References groups: ‘effort’, Over-commitment, E-R ratio: First Quartile; ‘reward’: Quartile 4. * Model 1. Adjusted by age. ** Model 2. Adjusted by age, gender, marital status, race/skin color, education.

**Table 4 ijerph-16-03025-t004:** Analysis de of the interaction of E–R ratio and ‘over-commitment’ with depressive episodes based on the multiplicative (product-term) criterion. ELSA-Brasil, 2012–2014.

Variable	PR	95%CI	*p* Value	PR *	95%CI	*p* Value
E-R ratio	1.71	1.30–2.25	<0.001	1.71	1.29–2.25	<0.001
‘Over-commitment’	1.79	1.36–2.36	<0.001	2.02	1.53–2.68	<0.001
Multiplicative term (E-R Ratio × OC)	1.06	0.71–1.58	0.752	1.06	0.71–1.58	0.769

* Adjusted by age, sex, marital status, race/skin color, education.

**Table 5 ijerph-16-03025-t005:** Analysis of interaction ER ratio and ‘over-commitment’ based on the additivity criterion. ELSA-Brasil, 2012–2014.

Variable	Prev. (%)	PR *	PR **	95%CI	Excess of Prevalence ^e^	Measure	95%CI
(1) E-R Ratio = 0, ‘over-commitment’=0	3.29	1.00	1.00	-	-		
(2) ER Ratio = 1, ‘over-commitment’=0	5.65	1.71	1.71	1.29–2.25	2.36		
(3) E-R Ratio = 0, ‘over-commitment’=1	5.89	1.79	2.02	1.53–2.68	2.60		
(4) E-R Ratio = 1, ‘over-commitment’ = 1	10.79	3.28	3.68	2.97–4.56	7.50		
Expected combined effect ^a^	8.25	2.01	2.73				
RERI ^b^						0.945	0.81–1.80
AP ^c^						0.256	0.04–0.46
S ^d^						1.543	1.00–2.39

Interaction exists if RERI ≠ 0 or AP ≠ 0 or S ≠ 1; *p* Value for RERI: 0.03; AP: 0.01; and S = 0.05. * PR without adjustment ** Adjusted for age, sex, marital status, race/skin color, education. ^a^ Expected combined effect: Prevalence rate = P01 − P00 + P10 − P00 + P00/Prevalence ratio: PR01 − PR00 + PR10 − PR00 + PR00. ^b^ Excess risk due to interaction (RERI) = PR_11_ − PR_01_ − PR_10_ + 1. ^c^ Attributable proportion due to interaction (AP) = (PR_11_ − PR_01_ − PR_10_ + 1)/PR_11._
^d^ Synergy index (S) = (PR_11_ − 1)/(PR_01_ + PR_10_ − 2). ^e^ Excess of prevalence ratio = (EP = Pexposure − Pno exposure).

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
