# Peer review of "Effort-Reward Imbalance, Over-Commitment and Depressive Episodes at Work: Evidence from the ELSA-Brasil Cohort Study"

_ijerph, 2019, doi:10.3390/ijerph16173025_

Round 1
Reviewer 1 Report
An excellently written article, and methodologically and statistically sound. It will contribute substantially to supporting the relevance of the ERI model and its relationship to mental health outcomes in developing countries.
Some minor comments:
Abstract line 22 - change to model - delete "s" from models
pg3 line 30 - I am not sure I understand why "physically demanding" was taken out considering the sample included nurses and maintenance and repair workers? Surely these workers may have physical demands?
pg 3 line 43 - The ERI ratio of over 1, indicating imbalance, is not actually used later in the statistical analyses? It seems the scores were ranked into quartiles (not using this "over 1" description). This could be a bit confusing to readers - as the article essentially defines those in the top quartile of ERI as "high" or imbalanced - not "over one"? Some small adjustments may make this more clear.
pg 4 line 23 - There was no age group categories under 40 (everyone under 40 was one category). The mean age was also quite high for the sample. Acknowledge in limitations?
p 8 line 3 - Begin sentence with A similar result (A missing)
Broader comments:
There are great gender (and regional) inequalities in educational attainment in Brazil (OECD, Education at a glance, 2018). Furthermore, other ERI research has found differences in relationships between ERI and health when analysing males and females separately. Perhaps justify your position on choosing to analyse the data as a group, rather than the option of reporting males and females separately? The sample is also not representative of typical tertiary attainment in Brazil (the sample seems very high % on this?) – the implications should be made a bit clearer (maybe limitation section).
Some ERI research articles use tertiles as opposed to quartiles. Is there a motivation for choosing one over the other?
Discussion section:
Dysfunctional reward processing is central to the pathophysiology of depression. Perhaps this link should be pointed out considering your strong findings on low rewards and depression. There certainly may be bi- directionality involved.
If there is space, it would be beneficial to give a practical example of an intervention to improve rewards in the workplace. This would help practitioners reading the article.
Reviewer 2 Report
This is a very clearly conceptualized and conducted study that provides a valuable contribution to the literature exploring mental health in the workplace by examining circumstance in a non-High Income Country setting.
The only conceptual concern that I have with the paper is that I do not think that the construct of "over-commitment" is adequately introduced. If this was added (i.e. even just applying the explanation that is explained in the 2016 Siegrist & Li paper (ref 15), I think this would help what is otherwise an appropriate analysis and discussion. Right now the reference at the first mention of over-commitment is the 1996 Siegrist paper which does not mention this - so ADDING the later Siegrist paper and having a bit more explanation (perhaps in an additional sentence) would be helpful.
As well, While the paper is well-organized and well written with respect to the content, there is room for some nuanced improvement in the English language terminology. I am providing some examples below of how revised wording can help (an attachment elaborating on this is also attached):
PAGE 1
Abstract :
Line 21 CHANGE models TO model
It is only the ERI model that is applied (the other model considered is not applied)
Line 23 CHANGE models’ to model’s
As singular
Line 27 CHANGE … prevalence ratio TO …prevalence ratio (PR)
The abbreviation is used below in presenting values, so it should be spelled out at first mention
LINES 27 – 30 CHANGE IN THIS WAY:
The quartiles of highest effort (PR=1.85;1.44-2.37), highest over-commitment (PR=3.62;2.80-4.70) and lowest reward (PR=3.44;2.55-4.64) were associated with DE, on adjusted models, as well was the E-R Ratio (PR=2.47;1.92-3.17).
The wording of highest is awkward – this would be clearer Ratio E-R is better expressed as E-R Ratio (appears a few times in the paper. g. line 30 ….. An additive interaction was identified between E-R Ratio ….
Introduction :
LINE 38 (* suggestion – the existing wording could be acceptable, but this has more nuance)
The nature of work and employment underwent significant changes
TO
The nature of work and employment HAS UNDERGONE significant changes
PAGE 2
LINE 3 (*) CHANGE confirmed TO have confirmed
LINE 6 (*) CHANGE were analyzed TO have been analyzed
LINES 38 & 39
Could add the abbreviations for Prevalence Ratios (PR) and Depressive Episodes (DE) in the paragraph introducing the hypotheses….
Page 5
Results
LINE 23
Change 14,014 to 14,014 participants
- you could use respondents but right now the term is missing
